# FiVA: Fine-grained Visual Attribute Dataset for Text-to-Image Diffusion Models

**Tong Wu**[1,2], **Yinghao Xu**[1], **Ryan Po**[1], **Mengchen Zhang**[3,5], **Guandao Yang**[1],
**Jiaqi Wang**[5], **Ziwei Liu**[4], **Dahua Lin**[2,5,6], **Gordon Wetzstein**[1]

[1] Stanford University   [2] The Chinese University of Hong Kong   [3] Zhejiang University
[4] S-Lab, NTU   [5] Shanghai Artificial Intelligence Laboratory   [6] CPII under InnoHK

## Abstract

Recent advances in text-to-image generation have enabled the creation of high-quality images with diverse applications. However, accurately describing desired visual attributes can be challenging, especially for non-experts in art and photography. An intuitive solution involves adopting favorable attributes from source images. Current methods attempt to distill identity and style from source images. However, "style" is a broad concept that includes texture, color, and artistic elements, but does not cover other important attributes like lighting and dynamics. Additionally, a simplified "style" adaptation prevents combining multiple attributes from different sources into one generated image. In this work, we formulate a more effective approach to decompose the aesthetics of a picture into specific visual attributes, letting users apply characteristics like lighting, texture, and dynamics from different images. To achieve this goal, we constructed the first fine-grained visual attributes dataset (FiVA) to the best of our knowledge. This FiVA dataset features a well-organized taxonomy for visual attributes and includes 1 M high-quality generated images with visual attribute annotations. Leveraging this dataset, we propose a fine-grained visual attributes adaptation framework (FiVA-Adapter) , which decouples and adapts visual attributes from one or more source images into a generated one. This approach enhances user-friendly customization, allowing users to selectively apply desired attributes to create images that meet their unique preferences and specific content requirements. The data and models will be released at `https://huggingface.co/datasets/FiVA/FiVA`.

## 1 Introduction

Imagine an artist drawing a picture; he or she not only exhibits unique styles, identities, and spatial structures but also frequently integrates personal elements such as brushstrokes, composition, and lighting effects into their creations. These detailed visual features capture profound personal emotions and artistic expressions. However, despite the capability of current text-to-image models to generate high-quality images from textual or visual prompts, they encounter substantial challenges in effectively controlling these fine-grained visual concepts, which vary widely across different artistic domains. This limits the practical applications of text-to-image models in various fields.

Recently, significant research efforts have been made to advance controllable image generation. In particular, numerous studies have explored the use of *personalization* techniques to preserve the identity of an object or person across different scenarios [23, 34]. On the other hand, there have been attempts to control the generation process by conditioning on the *style* and *spatial structure* of reference images [6, 28, 31, 7, 35], which involves mimicking abstract styles or leveraging spatial cues such as edge maps, semantic masks, or depth. Yet, these methods are limited to specific aspects and fall short in terms of generalizability. These methods use complex images as unified references

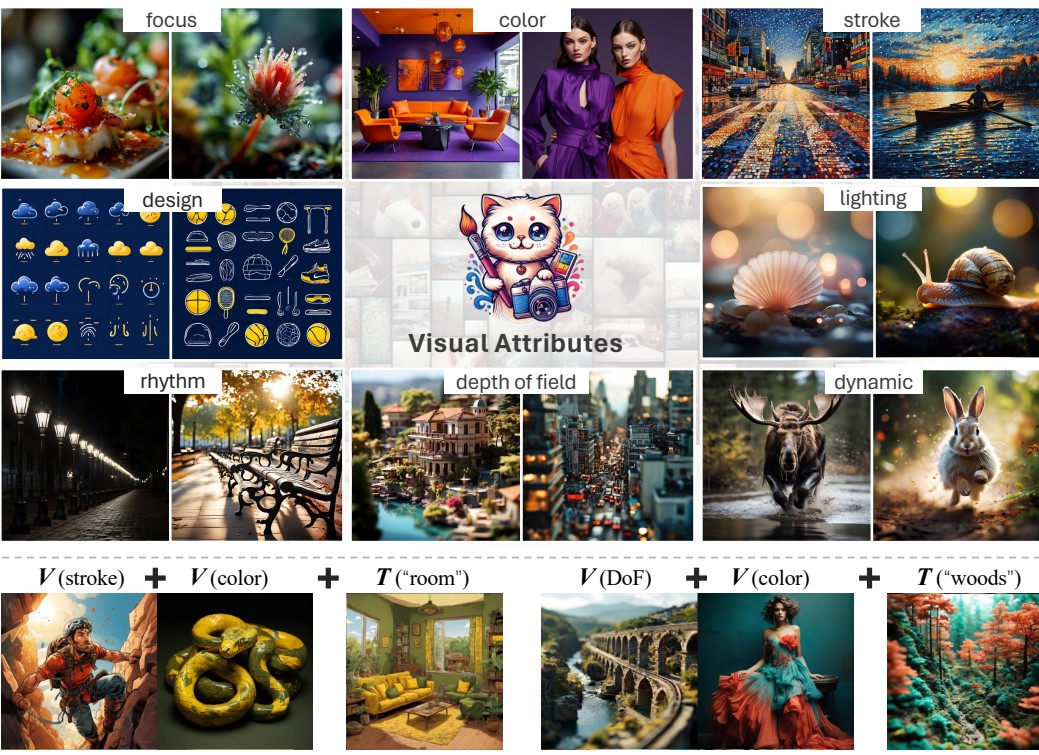

Figure 1: **Overview.** We propose the FiVA dataset and adapter to learn fine-grained visual attributes for better controllable image generation.

without adequately disentangling visual attributes, resulting in a lack of control over generation based on specific attributes of the conditional images. This underscores the importance of extracting fine-grained visual concepts from images to achieve controllable generation. However, achieving this requires a well-annotated image dataset that displays various fine-grained attributes within the images and a unified framework to adapt these different attributes to facilitate generation.

In this paper, we present a comprehensive fine-grained visual attributes dataset (FiVA), featuring image pairs that meticulously delineate a variety of visual attributes, as shown in Figure 1. Instead of annotating the real-world images, we propose to leverage advanced 2D generative models within an automated data generation pipeline for data collection. We develop a systematic approach that includes attribute and subject definition, prompt creation, LLM-based filtering, and human validation to construct a dataset annotated with diverse visual attributes across 1 M image pairs. Building on this dataset, we introduce a fine-grained visual attributes adaptation framework (FiVA-Adapter) designed to control the fine-grained visual attributes during the generation process. Specifically, we propose to integrate a multimodal encoder, Q-former, into the image feature encoder before its insertion into the cross-attention modules. This integration aids in understanding tags or brief instructions for extracting image information. During inference, our approach allows for the isolation of specific visual attributes from the reference image before applying them to a target subject, and even the combination of different attribute types. This capability enables free and diverse user choices, significantly enhancing the adaptability and applicability of our method.

We conduct extensive experiments across a variety of attribute types on both synthetic and real-world test sets. Our results demonstrate that our method outperforms baseline methods in terms of precise controllability in attribute extraction, high textual alignment regarding the target prompt, and the flexibility to combine different attributes. This work aims to cater to an increasingly diverse array of user needs, recognizing the rich informational content that visual media embodies. Our hope is that it will pave the way for innovative applications that harness the full potential of visual attributes in myriad contexts.

## 2 Related Words

**Image Generation with Diffusion Models** Diffusion models have seen significant advancements in the field of image generation, driven by their impressive generative capabilities. DDPM [8] employs an iterative denoising process to transform Gaussian noise into images. The Latent Diffusion Model [21] enhances traditional DDPMs by employing score-matching in the image's latent space and introducing cross-attention-based controls. Rectified flow [17] improves training stability and enhances image synthesis by introducing flow matching. Another line of research focuses on architectural variants of diffusion models. The Diffusion Transformer [18] and its variants [10, 2–4] replace the U-Net backbone with transformers to increase scalability on large datasets.

**Controllable Image Generation** The ambiguity of relying solely on text conditioning often results in weak control for image diffusion models. To enhance guidance, some works adopt the scene graph as an abstract condition signal to control the visual content. To allow for more spatial control, several attempts such as ControlNet [35] and IP-Adapter [34] propose to build adaptors for visual generation by incorporating additional encoding layers, thus facilitating controlled generation under various conditions such as pose, edge, and depth. Further research investigates image generation under specified image conditions. Techniques like ObjectStitch [29], Paint-by-Example [33], and AnyDoor [5] leverage the CLIP [20] model and propose diffusion-based image editing methods conditioned on images. AnimateAnyone [9] suggests using a reference network to incorporate skeletal structures for image animation. However, these methods primarily focus on the visual attributes like identity and spatial structure of reference images. Other works such as DreamBooth [23], CustomDiffusion [12], StyleDrop [27], and StyleAligned [7] approach controlled generation by enforcing consistent style between reference and generated images. Yet even among these methods, the definition of style is inconsistent. We propose a fine-grained visual attribute adapter that decouples and adapts visual attributes from one or more source images into generated images, enhancing the specificity and applicability of the generated content.

**Datasets for Image Generation** Datasets provide a foundation for the rapid progress of diffusion models in image generation. MS-COCO [16] and Visual Genome [11], built with human annotation, are limited in scale to 0.3 and 5 million samples, respectively. The YFCC-100M [30] dataset scales up further, containing 99 million images sourced from the web with user-generated metadata. However, the noisy labels from the web often result in text annotations that are unrelated to the actual image content. The CC3M [26] and its variant CC12M [1] utilize web-collected images and alt-text, undergoing additional cleaning to enhance data quality. LAION-5B [25] further scales up the dataset to 5 billion samples for the research community, aiding in the development of foundational models. DiffusionDB [32] leverages large-scale pre-trained image generation models to create a synthetic prompt-image dataset, containing imaged generated by Stable Diffusion using prompts and hyperparameters specified by real users. However, these datasets only provide a coarse description of the images. DreamBooth [23] and CustomDiffusion [12], techniques focused on text-to-image customization, also provide their own dataset, containing sets of images capturing the same concept. However, such datasets focus only on individual concepts, and are limited in sample volume. Thus, we propose the fine-grained visual attributes dataset (FiVA) with high-quality attribute annotations to facilitate precise control of generative models with specific attribute instructions from users.

## 3 Dataset Overview

Visual attributes encompass the distinctive features present in photography or art. However, there exists no dataset annotated with fine-grained visual attributes, making the fine-grained control in generative models inapplicable. In the following, we present the details for how to construct the FiVA dataset with fine-grained visual attributes annotation. We first introduce our taxonomy for the fine-grained visual attributes. We then illustrate how to create large-scale text prompts with diverse attributes, which are further used for generating image data pairs using state-of-the-art text-to-image models. Considering the imperfect text-image alignment of current generative models, a data filtering stage is implemented to further refine the precision. Additionally, we conduct thorough human validation to ensure the quality of the dataset. We perform a comprehensive statistical analysis of the compiled dataset in the supplementary material.

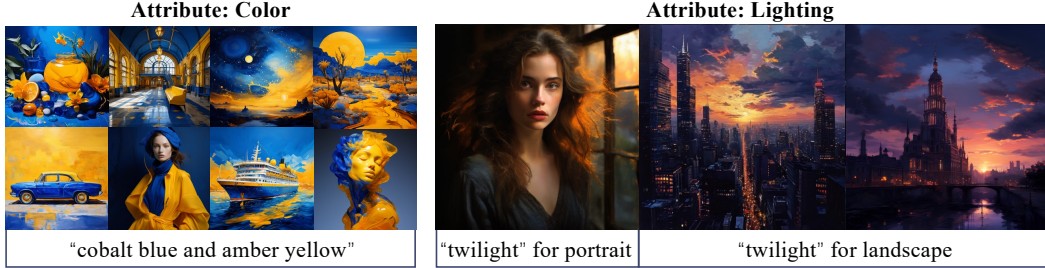

Attribute: Color — "cobalt blue and amber yellow" — *The attribute shares across all subjects.*

Attribute: Lighting — "twilight" for portrait — "twilight" for landscape — *The attribute is range-sensitive.*

Figure 2: **Examples of visual consistency application range.** Some visual attributes, such as 'color' and 'stroke,' are easily transferable across different subjects (**left**). However, other attributes, like 'lighting' and 'dynamics,' are range-sensitive, meaning they produce varying visual effects depending on the domain (**right**), resulting in more fine-grained, subject-specific definitions of sub-attributes.

Table 1: **Statistics and human validation results.** We report the number of individual images containing each specific visual attribute (some images could contain multiple attributes), along with human validation accuracy and cross-agreement measured by standard deviation.

| Attributes | Color | Stroke | Lighting | Dynamic | Focus_and_DoF | Design | Rhythm | Average |
|---|---|---|---|---|---|---|---|---|
| Number | 429K | 301K | 370K | 89K | 107K | 66K | 52K | 202K |
| Accuracy | 0.96 | 0.92 | 0.76 | 0.81 | 0.85 | 0.89 | 0.73 | 0.84 |
| Standard Deviation | 0.12 | 0.15 | 0.19 | 0.14 | 0.12 | 0.14 | 0.18 | 0.15 |

## 3.1 Data Construction Pipeline

**Taxonomy of Visual Attributes.** Visual attributes are a broad concept, varying significantly in different use cases. Therefore, we choose some of the most general types to cover a wide range of application scenarios. Specifically, we categorize these attributes into the following groups: `color, lighting, focus and depth of field, artistic stroke, dynamics, rhythm, and design`. Within each group, we further refine them into comprehensive subcategories. Initially, we reference examples from professional texts and then use GPT-4 to generate additional entries. We manually filter out any redundant or unreasonable suggestions. (Please refer to the supplementary material for more details). This comprehensive taxonomy enables us to encompass a wide range of applications in photography, art, and design, ensuring that our dataset is versatile and applicable across various visual domains.

**Prompts and Paired Images Generation.** We aim to construct image-image pairs with the same specific visual attribute. However, achieving this by filtering existing text-image datasets, LAION5B [24], etc. is very challenging due to the ambiguity in textual descriptions, which makes it difficult to accurately select the image pairs from the datasets. Therefore, we opt to construct the dataset using generative models: first, we generate prompts containing various visual attributes in bulk, and then we use state-of-the-art text-to-image models to generate images with these prompts.

For prompt generation, we augment the attribute names and descriptions with GPT-4 based on our taxonomy. Then, we combine each of the augment names or their combinations with specific objects to generate the final prompts. However, this process is non-trivial, as a specific branch of visual attributes may not be suitable for all kinds of random subjects. For example, "motion blur" can only be applied to dynamic subjects, and "candle light" cannot be applied to landscapes. We employ GPT-4 to construct a hierarchical subject tree, with parent nodes representing the primary categories of subjects and child nodes denoting specific objects within these main categories. We also utilize GPT-4 to associate visual attributes with the parent nodes in the subject tree to make the combination of them into a reasonable case. When creating prompts, we select $n$ (e.g., 1, 2, 3) attributes along with one associated subject according to the subject tree. We construct over 1.5 million initial prompts in total and used playground-v2.5 [14] to generate four images for each prompt. The generated images with the same attribute are considered to be positive pairs. However, we observe some disparity caused by the imperfect alignment with text as well as the different manifestations of the same

attribute on different subjects. A further cleaning and filtering process is therefore needed to enhance quality and precision.

**Range-sensitive Data Filtering.** Not all generated images with text prompts containing the same attribute exhibit similar visual effects. Figure 2 illustrates examples of varying application ranges for different attributes where visual consistency is present. To achieve attribute-consistent image pairs, we define specific ranges for each attribute within which two images are highly likely to maintain visual consistency. Utilizing our subject tree, we apply a *range-sensitive data filtering* approach hierarchically: if an attribute shows high visual consistency across subjects at the parent node, we record it as the range; if results are inconsistent, we proceed to its child nodes for further validation. The validation process involves constructing a $3 \times 3$ image grid generated with prompts containing the attribute and subjects sampled from a specific node. Using GPT-4, we assess image consistency for a given visual attribute by prompting: "I have a set of images and need to determine if they exhibit consistent <major attribute> traits of <specific attribute>." We ask GPT-4 to identify any inconsistent image IDs. This sampling is repeated multiple times, and if the proportion of inconsistent images remains below a predefined threshold, we consider the images consistent for the specified visual attribute within the given range.For attributes that cannot maintain consistency even at the leaf nodes, we simply remove them.

After filtering, we further compile statistics on the number of individual images for each major visual attribute, as shown in Table 1. Note that these numbers do not add up to the total image count, as each image can possess one or multiple different attributes.

## 3.2 Human Validation

After filtering the data, we randomly selected 1,400 image pairs (200 per attribute) for human validation. Each pair was evaluated by three of eight trained annotators for attribute similarity, with the final result determined by majority vote. These pairs were randomly matched based on shared attribute descriptions, and annotators focused solely on judging the visual similarity of each pair's attributes—alignment with the original text prompt was not required.

Table 1 displays the accuracy for each attribute type, along with the overall average, underscoring the dataset's high precision and robustness for this study. To assess annotator agreement, all eight annotators also evaluated a subset of 350 image pairs (50 per attribute). The table further includes the standard deviation of these assessments, demonstrating strong consistency among annotators.

## 4 Method

In the following, we will show how we leverage FiVA dataset to build a fine-grained visual attribute adapter (FiVA-Adapter) that enables controllable image generation using images as prompts for specific visual attributes.

## 4.1 Preliminary

**Image Prompt Adapter** (IP-Adapter) [34] is crafted to enable a pre-trained text-to-image diffusion model to generate images using image prompts. It comprises two main components: an image encoder for extracting image features, and adapted modules with decoupled cross-attention to embed these features into the pre-trained text-to-image diffusion model. The image encoder employs a pre-trained CLIP [20] image encoder followed by a compact trainable projection network, which projects the feature embedding into a sequence of features of length $N$. The projected image features are then injected into the pre-trained U-Net model [22] via decoupled cross-attention, where different cross-attention layers handle text and image features separately. The formulation can be defined as:

$$Z^{new} = \text{Softmax}\left(\frac{QK^T}{\sqrt{d}}\right)V + \text{Softmax}\left(\frac{Q(K')^T}{\sqrt{d}}\right)V', \tag{1}$$

where $Q = ZW_q, K = F_tW_k, V = F_tW_v, K' = F_vW'_k, V' = F_vW'_v$. $F_t$ and $F_v$ indicate the text and image condition features, respectively. $W'_k$ and $W'_v$ represent the weight matrices in the new cross-attention layer dedicated to the visual features. These matrices are the sole new parameters to

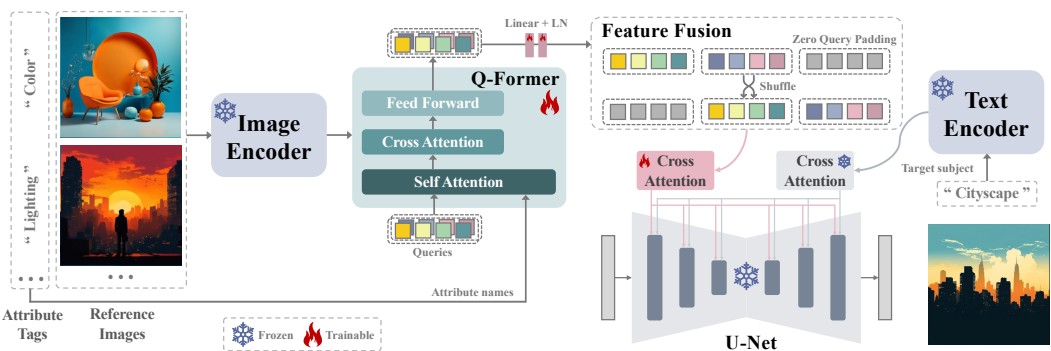

Figure 3: **FiVA-Adapter architecture and training pipeline.** FiVA-Adapter has two key designs: 1) Attribute-specific Visual Prompt Extractor, 2) Multi-image Dual Cross-Attention Module.

be trained within this module and are initialized from $W_k$ and $W_v$ to facilitate faster convergence. In this paper, we follow the decoupled cross-attention mechanism for adapting image prompts.

**Q-Former** [15] was initially introduced as a trainable module designed to bridge the representation gap between a image encoder and LLM. It comprises two transformer submodules: one functions as an image transformer to extract visual features, and the other serves as a text encoder and decoder. It takes learnable query tokens as input and associate with text via self-attention layers, and with frozen image features through cross-attention layers, ultimately producing text-aligned image features as output. Blip-Diffusion [13] utilizes Q-Former to derive text-aligned visual representations, allowing a diffusion model to generate new subject renditions from these representations. DEADiff [19] employs two Q-Formers, each focusing on "style" and "content" to extract distinct features for separate cross-attention layers, thereby enhancing the disentanglement of semantics and styles. It makes the Stylization Diffusion Model follow the 'style' of reference images better. In this work, we propose the FiVA-Adapter that focuses on fine-grained control of visual attributes. Thus, disentangling the 'content' and 'style' is not our goal.

### 4.2 Fine-grained Visual Attribute Adapter

Our goal is to generate the target image I following the attribute instruction of visual prompts and the text condition. Specifically, the generation can be formulated as:

$$\mathbf{I} = \mathrm{G}(\mathcal{V}, \mathcal{A}, y), \tag{2}$$

where $\mathrm{G}(\cdot)$ is the generator, $\mathcal{V} = \{\mathbf{V}_k | k \in [1, N]\}$ represents the visual prompts, $\mathcal{A} = \{a_k | k \in [1, N]\}$ denotes the corresponding attribute instructions, and $y$ is the text prompts.

To achieve our goal, our framework is composed of two key components: 1) *Attribute-specific Visual Prompt Extractor*: A feature extractor that can extract the attribute-specific image condition feature $\mathbf{F}_k$ from image $\mathbf{V}_k$ with respect to $a_k$. 2) *Multi-image Dual Cross-Attention Module*: A module can embed both text prompt conditions and multiple image conditions into U-Net with two dedicated Cross-Attention Modules for multi-conditional image generation.

The overall pipeline of our method is illustrated in Figure 3.

**Attribute-specific Visual Prompt Extractor.** Inspired by BLIP-Diffusion [13], we employ the Q-former module to extract the attribute-specific image condition features. Specifically, the Q-former takes both image $\mathbf{V}_k$ and attribute instruction $a_k$ as inputs. It models the semantic relationship between the image and the attribute instruction, expected to extract condition feature that aligned with the given attribute. The extracted feature is then further projected by a projector comprises a Linear Layer and LayerNorm into the image condition $\mathbf{F}_k$ to meet the input channel of the following cross-attention module.

**Multi-image Dual Cross-Attention Module.** IP-Adapter [34] introduces a Dual Cross-Attention Module where one cross-attention for text prompt condition and the other one for a single-image condition. In our work, we extend this module to adapt to multi-image conditions, facilitating multi-image controls. Specifically, we set a fixed number of attributes $N$ to control the generated

|  | Reference | **Ours** | DB_Lora | IP-Adapter | DEADiff | StyleAlign |

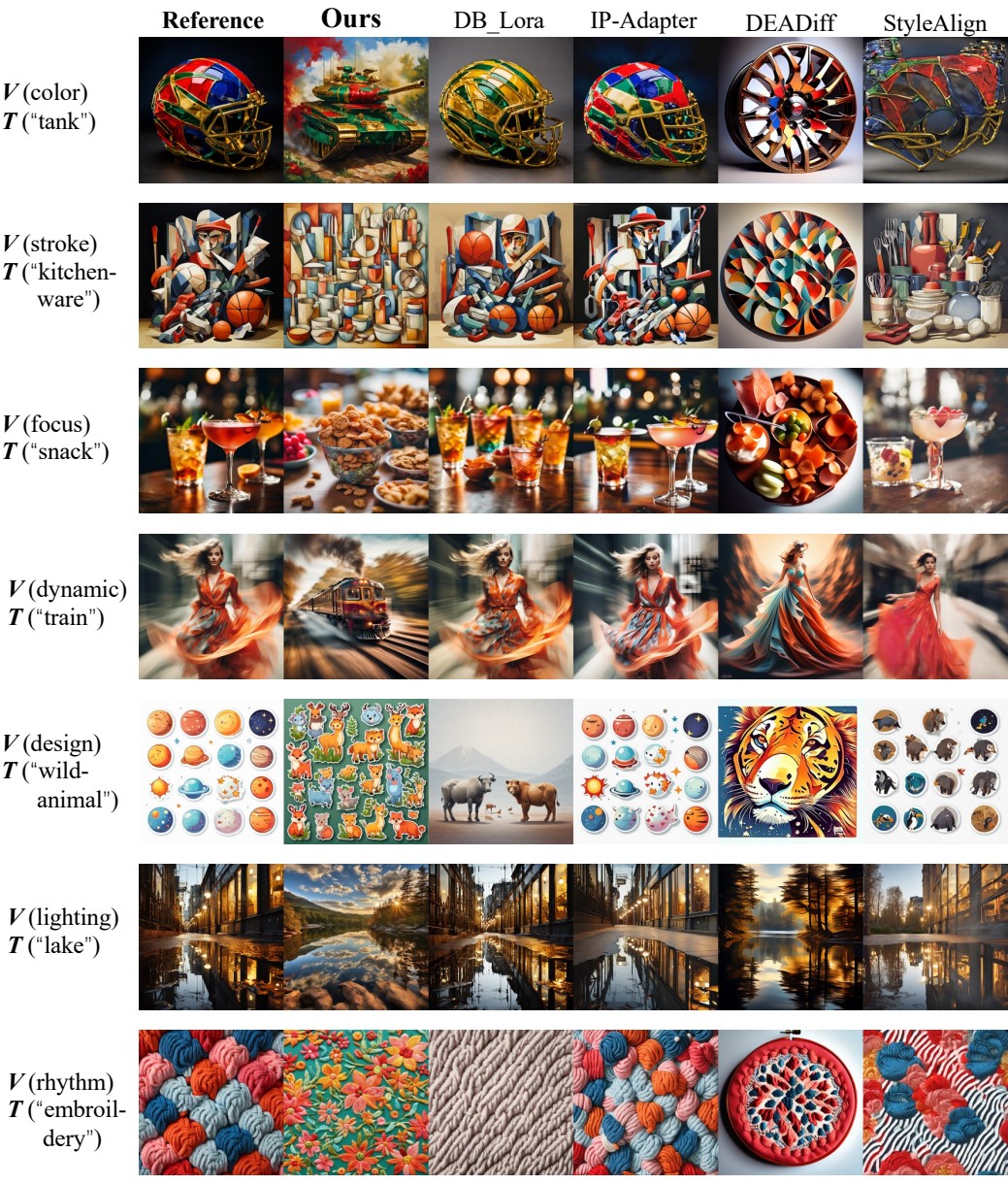

$V$(color)
$T$("tank")

$V$(stroke)
$T$("kitchen-ware")

$V$(focus)
$T$("snack")

$V$(dynamic)
$T$("train")

$V$(design)
$T$("wild-animal")

$V$(lighting)
$T$("lake")

$V$(rhythm)
$T$("embroil-dery")

Figure 4: **Qualitative comparisons on single attribute transferring.**

image simultaneously. This allows us to use a fixed number of tokens as input for the cross-attention. The image condition features $\mathbf{F}_k$ prepared by the Q-Former and channel projector are concatenated into $\mathcal{F} = [\mathbf{F}_0, \mathbf{F}_1, \ldots, \mathbf{F}_N]$. Notably, since not all images in our dataset have $N$ attributes, for a target image I with fewer than $N$ attributes, we use unconditional image features $\mathbf{F}^{zero}$ to pad the sequence. The unconditional feature $\mathbf{F}^{zero}$ is generated by feeding a zero-image and an empty text into the Q-Former and followed channel projector. To ensure the condition signals are invariant to the order of prompts, we randomly shuffle the multi-image conditions during training. Finally, the multi-image conditions are passed to the cross-attention module as described in Equation 1.

**Image Prompts and Attribute Sampling .** For a target image characterized by a series of visual attributes and a specific subject, we randomly select reference images from the training set that share the same attribute. We incorporate LLM-based data filtering to enhance data sampling. It ensures that the subjects of sampled image prompts can share the attribute with the target image's subject.

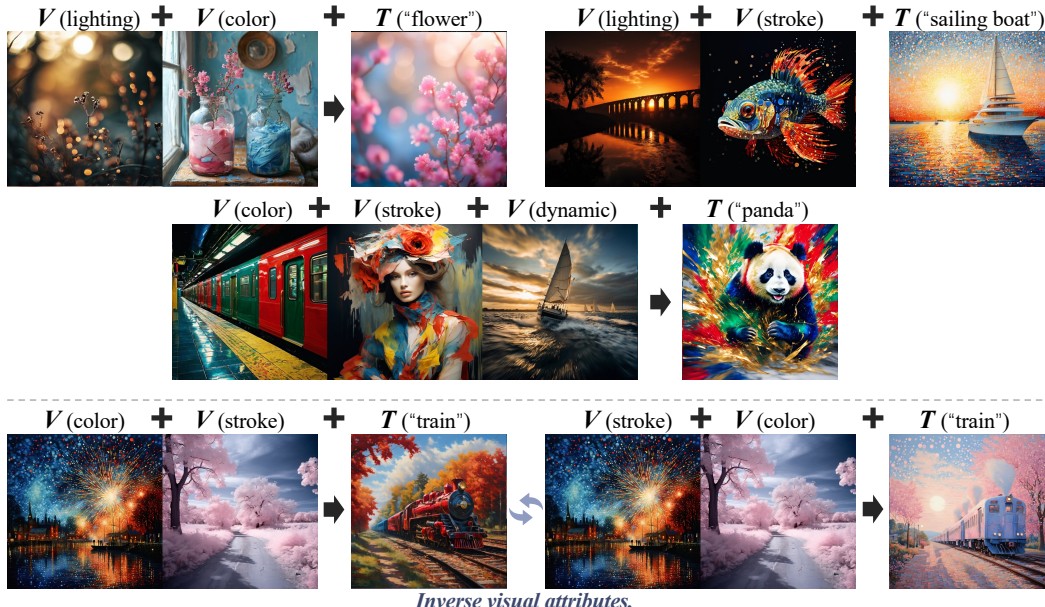

Figure 5: **The combination of multiple visual attributes** enables the integration of specific characteristics from different reference images into the target subject.

Table 2: **User Study and CLIP Scores.** Both quantitative results demonstrate our superior performance over the baseline in terms of both subject and joint accuracy.

| Metrics | | DB-Lora | IP-Adapter | DEADiff | StyleAligned | **Ours** |
|---|---|---|---|---|---|---|
| User-Study | Sub-Acc | 0.393 | 0.163 | 0.605 | 0.520 | **0.817** |
| | Attr&Sub-Acc | 0.240 | 0.150 | 0.260 | 0.298 | **0.348** |
| CLIP-Score | in-domain | 0.180 | 0.161 | 0.211 | 0.196 | **0.228** |
| | out-domain | 0.177 | 0.135 | 0.205 | 0.189 | **0.229** |

Additionally, we enhance the flexibility of attribute instructions by implementing tag augmentation on the attribute text, broadening the range of instructional keywords to accommodate various user inputs. Specifically, we prepare a list of augmented tags for each attribute using GPT-4. The augmented attribute text for prompt images is then randomly sampled from this candidate list of tags.

**Training and Inference** The training and inference setting of our framework are similar with the IP-Adapter [34]. The learning rate is set to 2e-5 and weight decay is set to 1e-3 for stabling the training. The Q-former, channel projector, and multi-image cross-attention are trained and other parameters are frozen.

## 5 Experiment

### 5.1 Experimental Settings

**Baselines** We compare our method with the state-of-the-art methods for customized image generation, including Dreambooth-Lora [23], IP-Adapter [34], DEADiff [19], and StyleAligned [7]. Please find the implementation details of both our method and the baselines in the supplementary material.

**Evaluation Metrics** In order to evaluate the results systematically, we propose a small validation set with 100 reference images covering the seven attributes, together with 4 target subjects for each of them. The target subjects include 2 in-domain ones, which are randomly selected from the same major category with the subject from the reference image, and 2 out-domain ones, which are randomly selected from the other major categories. For the evaluation, we focus on two aspects: *attribute-*

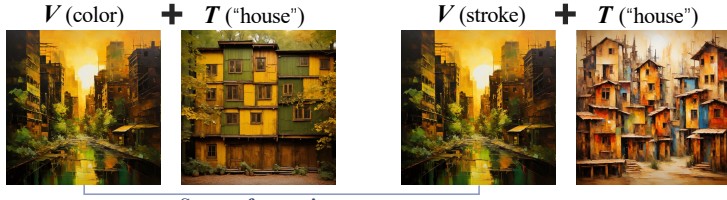

*V* (color)  **+**  *T* ("house")   *V* (stroke)  **+**  *T* ("house")

Same reference image

Figure 6: **Attribute decomposition.** One reference image can be decomposed into different attributes via different tags.

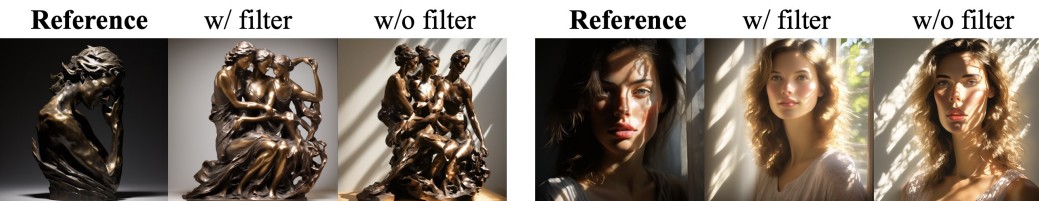

**Reference**   w/ filter   w/o filter   **Reference**   w/ filter   w/o filter

Figure 7: **Ablation on range-sensitive data filter.** It helps improve the attribute accuracy and protect the original generation capacity.

*accuracy* and *subject-accuracy*. Given the subjectivity of the issue and the lack of quantitative metrics, we incorporate both user studies and GPT scoring on our proposed validation set. Specifically, both users and GPT are instructed as follows: subject-accuracy is determined by whether the subject in the generated image is correct, while attribute-accuracy is evaluated in conjunction with the subject, referred to as attr&sub-accuracy, to eliminate the "image variance" solution that might artificially inflate accuracy. Please refer to the supplementary materials for more details about the validation set, metric implementation, and GPT scoring results.

## 5.2 Quantitative Evaluation

We show quantitative evaluation results in Table 2, where both the CLIP-Score and user study results indicate that our method benefits from a higher subject accuracy. The user study further demonstrates that our method also excels in the joint accuracy of both subject and attribute. However, it is important to note that the joint accuracy is not high overall due to the challenging nature of this new task.

## 5.3 Qualitative Evaluation

**Comparisons with previous methods.** Figure 4 demonstrates that our method effectively transfers the specific attribute to the target. In contrast, other methods exhibit various issues. The DB-Lora and IP-Adapter usually tend to create image variations without accurately following the target subject. While DEADiff follows the target subject better, its attribute accuracy is poor, and often resulting in round images. Additionally, the StyleAligned model produces unstable image quality, with a high risk of generating anomalous images.

**Visual attributes decomposition from the reference image.** Based on our method, the visual information extracted from the same reference image can be different, depending on the tag input. Figure 6 shows an example of this, demonstrating the effectiveness of fine-grained attribute decomposition.

**Combination of multiple visual attributes into the target subject.** We demonstrate how multiple reference images can be used to combine specific attributes from each into a new image with a target subject. As shown in Figure 5, our method accurately extracts the specified attribute from two or even three reference images and combines them with the target subject. To further validate the method's sensitivity to the input tags, we conducted experiments by exchanging the tags for the same reference images. The results confirm that the visual attribute extraction is both valid and distinct for different tag inputs, highlighting the method's flexibility.

**Effect of range-sensitive data filter.** The filter introduced in Section 3.1 is aimed for regularizing the similarity application range of some attributes like lighting and dynamic modelling. In Figure 7, we show some qualitative results of its effects. First, it slightly enhances the attribute accuracy rate regarding the difficult visual attributes like lighting. What's more important, it also helps protect the generation capacity of the original pre-trained model from being harmed by the heavy noise in the paired data without filtering, leading to highly image quality and lower rate in producing failure cases like distorted human face and body.

## 6    Conclusion

In conclusion, our work addresses the limitations of current text-to-image models in controlling fine-grained visual concepts by introducing a comprehensive dataset with fine-grained visual attributes, FiVA dataset, and a novel visual prompt adapter along with it. Our approach enables precise manipulation of specific visual attributes, offering greater flexibility and applicability across various photography, artistic, and other practical domains. We believe that our contributions will pave the way for more sophisticated and user-driven image generation technologies. We will discuss the limitations and future works in the supplementary material.

**Limitations and Future Works.**  The main limitation of the dataset is its heavy reliance on the capacity of the generative model, which might constrain the realism, range of available visual attributes, and attribute accuracy between paired data. For example, specific attributes like photographic composition techniques or creative photography can hardly be created in this way. This might also introduce some bias in appearance distribution introduced by the generative model. In the future, we will consider collecting some high-quality data from platforms with professional photographers and designers, and involve human annotation to create paired data, which can further enhance the dataset with a more realistic data distribution and more complex visual attributes.

## 7    Acknowledgment

This study is supported by Shanghai Artificial lntelligence Laboratory, the National Key R&D Program of China (2022ZD0160201), the Centre for Perceptual and Interactive Intelligence (CPII) Ltd under the Innovation and Technology Commission (ITC)'s InnoHK. Dahua Lin is a PI of CPII under the InnoHK. It is also supported by the Ministry of Education, Singapore, under its MOE AcRF Tier 2 (MOET2EP20221-0012), NTU NAP, and under the RIE2020 Industry Alignment Fund – Industry Collaboration Projects (IAF-ICP) Funding Initiative, as well as cash and in-kind contribution from the industry partner(s).

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

In the supplementary material, we include links to the dataset, metadata, and documentation in Section A. We then introduce additional details on dataset construction and statistics in Section B. Further, we present more details on the experimental setup and additional experimental results in Section C. Finally, please also find the datasheet for the dataset in Section D.

# A  Dataset Information

## A.1  Dataset Link and Documentation

Our dataset, metadata, and its license are currently maintained on huggingface [1] for users to download: https://huggingface.co/datasets/FiVA/FiVA. It contains the generated images and their metadata, the the original taxonomy of visual attributes and subjects to create the prompts, and the data filtering file. For each of the images, the main visual attribute type, keyword, subject, and prompt is stored in the metadata. A detailed documentation of dataset structure and usage as well as an example of the metadata can be found in the dataset card via the URL above. The Croissant link can be find here https://huggingface.co/api/datasets/FiVA/FiVA/croissant.

## A.2  Author Statement and Data License

The authors bear all responsibility in case of violation of rights and confirm that this dataset is open-sourced under the Playground v2.5 Community License license. We will be adding more generated images from other generative models and release them under the corresponding licenses.

# B  Additional Details on Dataset Construction

**Details on attribute taxonomy and statistics.** When constructing the attribute library, for `color`, `lighting`, `dynamics`, `artistic stroke`, and `focus and depth of field`, we create a list of short descriptions or keywords for each kind of subcategory together with a list of major subjects that can fit into the description. When constructing the prompt, we simply link the attribute and the subject with a comma. For two specific attribute types, namely `rhythm` and `design`, the visual results can hardly be presented simply via short descriptions or keywords. We use long descriptions with "[sks]" denoting the placeholder for subjects that might fit into the sentence. Prompts are created by replacing "[sks]" with each of the subject candidates. We show the visualization of a rough distribution of attributes and subjects in Figure S2, as well as an example of constructing a pair of images that share similar lighting conditions. We also show some more examples of images with different visual attributes in Figure S1.

**Details on the Range-sensitive Data Filtering.** To achieve attribute-consistent image pairs, we need to establish a set of ranges for each attribute where any two images maintain consistency. We organize images into a hierarchy of `Set/Major-subject/Sub-subject`, with the largest set being the aforementioned "group of suitable subjects." Figure S3a shows an example of the hierarchical structure of images related to the attribute 'lighting: moonlight' featuring 7 major-subjects and over 100 sub-subjects. Within this hierarchy, each sub-subject corresponds to a list of images, where each image belongs to that sub-subject and possesses the visual attribute of 'lighting: moonlight'.

We apply *Range-sensitive Data Filtering* to this hierarchy: We first validate the consistency within each specific `Major-subject`. Subsequently, we validate the `Set` encompassing all validated major-subjects. For any major-subject that failed to pass the validation, we then check their `Sub-subjects`.

As shown in Figure S3b, from the range we want to verify, we sample 9 images and arrange them in a grid. Using GPT-4V, we assessed image consistency for a specific visual attribute. In our example, <major attribute> is *lighting*, and <specific attribute> is *moonlight*. For each range we want to verify, this sampling is repeated multiple times. If the mean proportion of inconsistent images remains below a predefined threshold of 0.1, we consider the images consistent for the selected visual attribute within the specified range.

---

[1] https://huggingface.co/

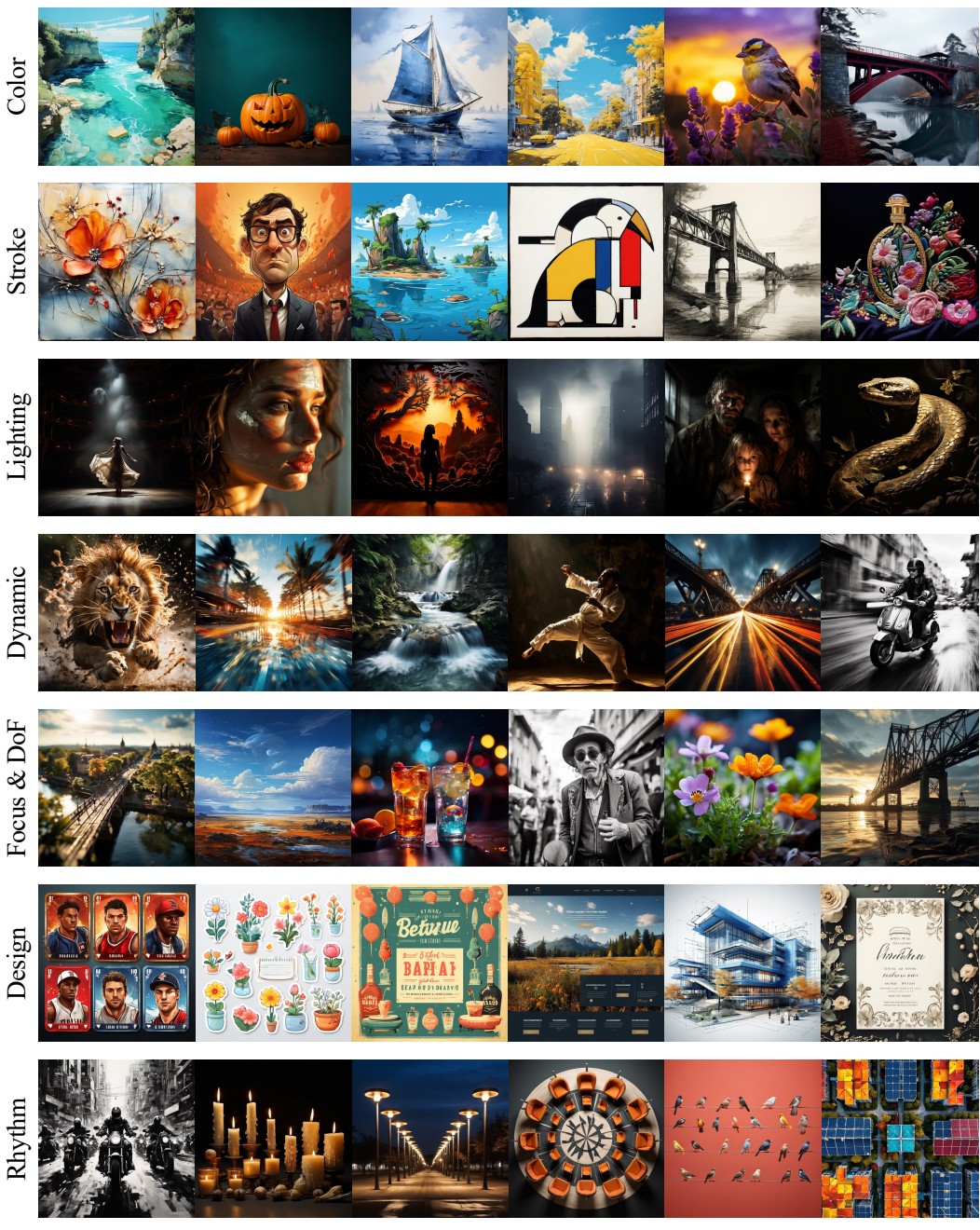

Figure S1: **More image examples with different visual attributes.**

## C  Additional Experimental Details

### C.1  Details on Experimental Setup

**Implementation Details.** For our methods, our framework's training and inference setting are similar to the IP-Adapter [34]. The learning rate is set to 2e-5, and weight decay is set to 1e-3 for stabilizing the training. The Q-former, channel projector, and multi-image cross-attention are trained, and other parameters are frozen. The training images are resized to $512 \times 512$. The model is trained for three epochs with the randomly shuffled training dataset. For each target image, the attribute images are randomly sampled.

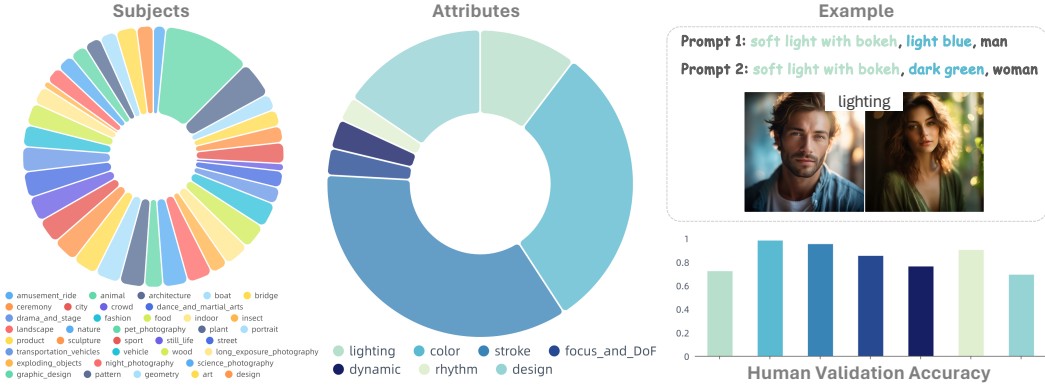

Figure S2: **Statistics and Analysis.** We visualize the rough distribution of visual attributes and subjects on the left. On the right, we show an example pair of images that shares similar lighting condition. We also visualize the attribute alignment accuracy via human validation here.

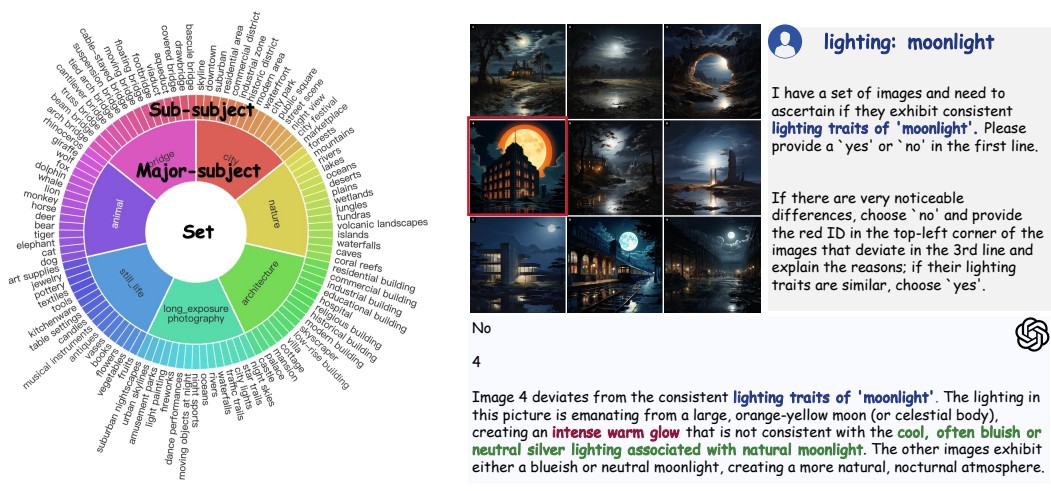

(a) **Subject Tree of lighting: moonlight.**  (b) **GPT4V based Range-sensitive Data Filtering.**

Figure S3: **Range-sensitive Data Filtering.** Taking the attribute *lighting: moonlight* as an example, **(a)** demonstrates the hierarchy of `Set/Major-subject/Sub-subject`. It lists the "group of suitable subjects" chosen for generating images with the visual attribute *lighting: moonlight*, along with sub-subjects under each major-subject. The "group of suitable subjects" refers to the pre-defined major-subjects that are applicable for the attribute. Due to space limitations, only 15 sub-subjects are listed for each major-subject. **(b)** verifies whether the images under *major-subject: architecture* exhibit consistent *lighting* traits of *moonlight*. The result shows that Image 4 exhibits inconsistencies, with the reasons provided.

For the baseline methods, we adopt the official code base and hyper-parameters for IP-Adapter [34], DEADiff [19], and Style-Aligned [7], and we use the implementation in diffusers [2] for Dreambooth-Lora [23] with only the reference image as training source.

**Details on Evaluation** The validation set for the user study contains 100 reference images with different visual attribute types. The distribution of the validation set reflects the inherent diversity of each attribute. We involve three times more data for the GPT study under the same distribution, thanks to its ability to scale up.

For the CLIP-Score, we use `ViT-L-14` model, and report the cosine similarity between the text feature of the target subject and the image feature of the generated image. For the user study,

---

[2]https://github.com/huggingface/diffusers

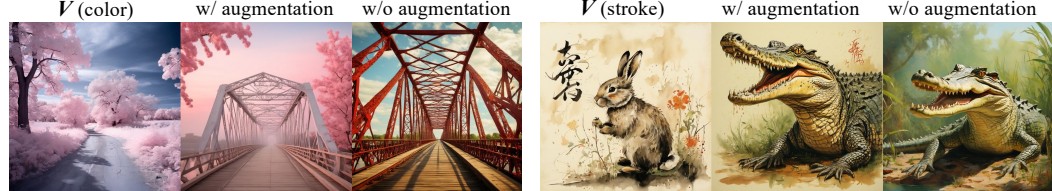

Figure S4: **Ablation on attribute input augmentation.** Models trained with tag augmentation handle slight deviations in input text during inference, while those without augmentation would fail in these cases.

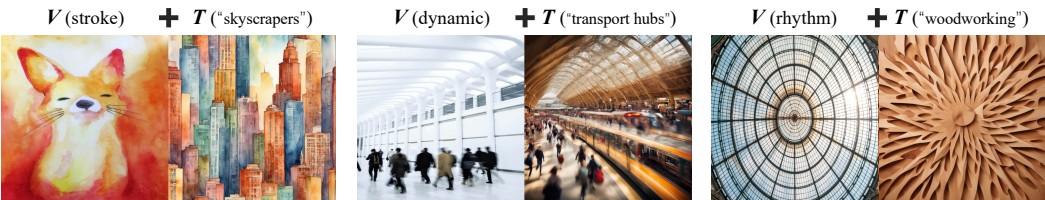

Figure S5: **Examples with real-world images.** We demonstrate that our adapter can be effectively extended to real-world images, which have a different distribution from generated images.

we send questionnaires to 30 volunteers with randomly shuffled image options. We are using the `gpt-4-turbo-2024-04-09` model for GPT-4V API inference. Detailed instructions for GPT-4V can be find in Figure S6.

### C.2  More Results

**GPT Study Results** Multi-modal Large Language Models (*e.g.*, GPT-4V(ision)) can offer a more scalable alternative to user studies, providing comprehensive analysis and judgment simultaneously. Specifically, we instruct the GPT-4V model to complete similar questionnaires as in the user study. An example of the instruction and GPT's output can be found in Figure S6. The GPT study results, shown in Table R1, demonstrate that our method outperforms the baselines in most attributes. However, the results for `Design` and `Rhythm` are not as strong, possibly due to the relatively small data scale for these two attributes.

**Effect of the input attribute augmentation.** During inference, users may present visual information in various ways. For example, "color" might be referred to as "hue" or "palette," and "dynamic" as "motion capture" or "action shot." Therefore, we add attribute name augmentation during Q-former training to accommodate diverse user inputs. As shown in Figure S4, when the input text slightly differs from the standard attribute names during inference, models trained with tag augmentation can still perform effectively, whereas those without augmentation fail to do so.

**Results on real-world data.** We show the generalization ability of the model to some real-world data collected from Unsplash [3] to verify the model's generation ability to some attributes beyond the training set. Results in Figure S5 shows that our adapter can be effectively extended to real-world images, which have a different distribution than generated images.

---

[3]https://unsplash.com/

Table R1: **GPT study results on each attribute type.** The Attr&Sub-Acc here denotes the accuracy when both the attribute transferring and target subject are correct.

| Methods | Attr&Sub-Acc | | | | | | | |
| | Color | Stroke | Lighting | Focus&DoF | Dynamic | Design | Rhythm | **Average** |
|---|---|---|---|---|---|---|---|---|
| DB-Lora | 0.516 | 0.478 | 0.358 | 0.485 | 0.480 | 0.600 | **0.607** | 0.503 |
| IP-Adapter | 0.323 | 0.403 | 0.340 | 0.364 | 0.520 | 0.440 | 0.500 | 0.413 |
| DEADiff | 0.161 | 0.209 | 0.245 | 0.485 | 0.400 | 0.080 | 0.357 | 0.277 |
| Style-Aligned | 0.581 | 0.552 | **0.396** | 0.606 | **0.600** | **0.660** | 0.571 | 0.567 |
| **Ours** | **0.780** | **0.647** | **0.396** | **0.727** | 0.560 | 0.510 | 0.521 | **0.592** |

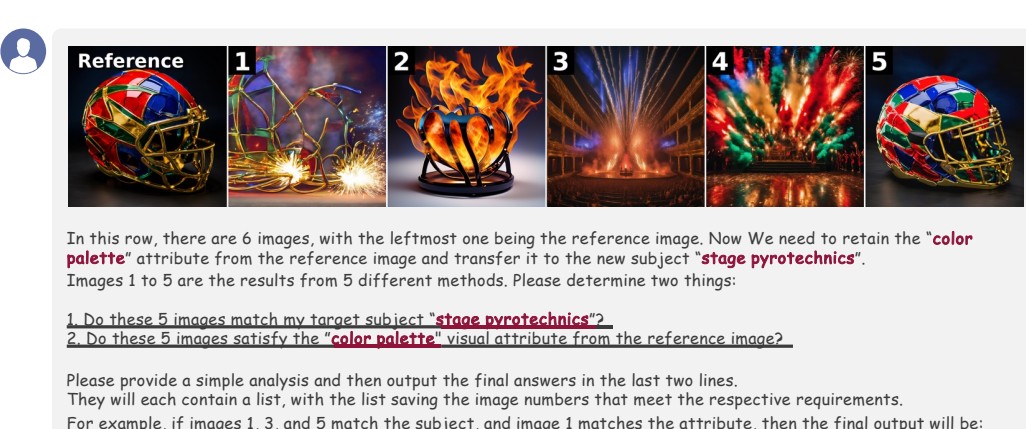

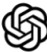

In this row, there are 6 images, with the leftmost one being the reference image. Now We need to retain the "**color palette**" attribute from the reference image and transfer it to the new subject "**stage pyrotechnics**".
Images 1 to 5 are the results from 5 different methods. Please determine two things:

1. Do these 5 images match my target subject "**stage pyrotechnics**"?
2. Do these 5 images satisfy the "**color palette**" visual attribute from the reference image?

Please provide a simple analysis and then output the final answers in the last two lines.
They will each contain a list, with the list saving the image numbers that meet the respective requirements.
For example, if images 1, 3, and 5 match the subject, and image 1 matches the attribute, then the final output will be:

**# Example**
**[2,3,5]**
**[3]**

THE LAST TWO LINES SHOULD ONLY CONTAIN THESE TWO LISTS!!!

1. Image 1 contains fireworks integrated within a frame structure, not matching the pyrotechnic display often associated with stage pyrotechnics.
2. Image 2 depicts a fiery display within a heart-shaped structure, somewhat representing a pyrotechnic element but still not typical of a staged event.
3. Image 3 clearly shows a stage set with a pyrotechnics display, fitting the "stage pyrotechnics" description accurately.
4. Image 4 again features a stage with vibrant fireworks, closely matching what you'd expect from "stage pyrotechnics."
5. Image 5, while beautifully matching the helmet in the reference in terms of subjects, does not relate to "stage pyrotechnics."

Regarding the "same color palette" attribute from the reference image:
- The reference image features a dynamic and vivid color range with predominant reds, blues, greens, and touches of yellow.
1. Image 1 attempts to incorporate similar vibrant colors but the colors are less vivid and more subdued compared to the reference.
2. Image 2 features intense orange and yellow tones, lacking the broader color variety of the reference.
3. Image 3, while mostly featuring golden and warm tones, lacks the varied color palette of the reference.
4. Image 4 infuses a rich variety of colors, aligning well with the diverse palette of the reference.
5. Image 5 also diligently replicates the reference's color palette with the same rich, vivid colors.

Conclusively:

[3,4]
[4,5]

Figure S6: **An example of the input, instruction, and output of the GPT study.** GPT-4V shows sufficient ability in understanding the problem and providing comprehensive analysis and judgement to these questions that can hardly be evaluated by other pre-trained models.

# D  Datasheet for Datasets

The following section contains answers to questions listed in datasheets for the dataset.

## D.1  Motivation

- For what purpose was the dataset created?
  The FiVA dataset is designed to facilitate research in fine-grained visual attributes, enabling user-friendly customization. It allows users to selectively apply desired attributes to create images that match their unique preferences and specific content requirements.

- Who created the dataset (e.g., which team, research group) and on behalf of which entity (e.g., company, institution, organization)?
  The authors of this paper.

- Who funded the creation of the dataset? If there is an associated grant, please provide the name of the grantor and the grant name and number.
  Stanford, The Chinese University of Hong Kong, Shanghai AI Lab, NTU, and CPII supported this work.

## D.2  Composition

- What do the instances that comprise the dataset represent (e.g., documents, photos, people, countries)?
  The FiVA dataset consists of a number of pairs of images that share similar visual attributes and corresponding meta data like attribute type and subject.

- How many instances are there in total (of each type, if appropriate)?
  The FiVA dataset contains 1M images generated by Playground-V2.5. More examples from other generative models are planned to be further added.

- Does the dataset contain all possible instances or is it a sample (not necessarily random) of instances from a larger set?
  The FiVA dataset is a new dataset generated using existing 2D generative models.

- What data does each instance consist of?
  Each instance contains an image with a prominent visual feature, such as color, stroke, lighting, and so on.

- Is there a label or target associated with each instance?
  Yes.

- Is any information missing from individual instances? If so, please provide a description, explaining why this information is missing (e.g., because it was unavailable). This does not include intentionally removed information, but might include, e.g., redacted text.
  N/A.

- Are relationships between individual instances made explicit (e.g., users' movie ratings, social network links)?
  N/A.

- Are there recommended data splits (e.g., training, development/validation, testing)?
  Yes. We provide a small subset for validation.

- Are there any errors, sources of noise, or redundancies in the dataset?
  Yes.

- Is the dataset self-contained, or does it link to or otherwise rely on external resources (e.g., websites, tweets, other datasets)?
  The dataset is self-contained.

- Does the dataset contain data that might be considered confidential (e.g., data that is protected by legal privilege or by doctor– patient confidentiality, data that includes the content of individuals' non-public communications)?
  N/A.

- Does the dataset contain data that, if viewed directly, might be offensive, insulting, threatening, or might otherwise cause anxiety?
  N/A.

- Does the dataset relate to people?
  Yes.

- Does the dataset identify any subpopulations (e.g., by age, gender)?
  N/A.

- Is it possible to identify individuals (i.e., one or more natural persons), either directly or indirectly (i.e., in combination with other data) from the dataset?
  N/A.

- Does the dataset contain data that might be considered sensitive in any way (e.g., data that reveals race or ethnic origins, sexual orientations, religious beliefs, political opinions or union memberships, or locations; financial or health data; biometric or genetic data; forms of government identification, such as social security numbers; criminal history)?
  N/A.

## D.3  Collection Process

- How was the data associated with each instance acquired?
  We used the open-source 2D generative model, Playground-V2.5 [14] to generate the dataset.

- What mechanisms or procedures were used to collect the data (e.g., hardware apparatuses or sensors, manual human curation, software programs, software APIs)?
  We develop an attribute library and subject tree to create the prompts, generate the images, and develop a range-sensitive filtering to enhance the pair-wise attribute alignment. We also perform human validation to verify the accuracy of the attribute alignment.

- If the dataset is a sample from a larger set, what was the sampling strategy (e.g., deterministic, probabilistic with specific sampling probabilities)?
  N/A.

- Who was involved in the data collection process (e.g., students, crowdworkers, contractors) and how were they compensated (e.g., how much were crowdworkers paid)?
  The authors of the paper participated in the data collection and verification process.

- Over what timeframe was the data collected?
  The data was collected during April and May of 2024.

- Were any ethical review processes conducted (e.g., by an institutional review board)?
  N/A.

- Does the dataset relate to people?
  Yes.

- Did you collect the data from the individuals in question directly, or obtain it via third parties or other sources (e.g., websites)?
  We generated the image data.

- Were the individuals in question notified about the data collection?
  The data is not collected from individuals.

- Did the individuals in question consent to the collection and use of their data?
  The data is not collected from individuals.

- If consent was obtained, were the consenting individuals provided with a mechanism to revoke their consent in the future or for certain uses?
  N/A.

- Has an analysis of the potential impact of the dataset and its use on data subjects (e.g., a data protection impact analysis) been conducted?
  Yes.

## D.4  Preprocessing/cleaning/labeling

- Was any preprocessing/cleaning/labeling of the data done (e.g., discretization or bucketing, tokenization, part-of-speech tagging, SIFT feature extraction, removal of instances, processing of missing values)?
  Yes. We provide a data filter.

- Was the "raw" data saved in addition to the preprocessed/cleaned/labeled data (e.g., to support unanticipated future uses)?
  Yes.

- Is the software that was used to preprocess/clean/label the data available?
  Yes, we use Python to preprocess/clean/label the data.

## D.5 Uses

- Has the dataset been used for any tasks already?
  Yes, for customized image generation.

- Is there a repository that links to any or all papers or systems that use the dataset?
  No.

- What (other) tasks could the dataset be used for?
  High-level perception tasks like aesthetic analysis.

- Is there anything about the composition of the dataset or the way it was collected and preprocessed/cleaned/labeled that might impact future uses?
  N/A.

- Are there tasks for which the dataset should not be used?
  N/A.

## D.6 Distribution

- Will the dataset be distributed to third parties outside of the entity (e.g., company, institution, organization) on behalf of which the dataset was created?
  No.

- How will the dataset will be distributed (e.g., tarball on website, API, GitHub)?
  The dataset are released on Huggingface: https://huggingface.co/datasets/FiVA/FiVA/.

- When will the dataset be distributed?
  The dataset will be gradually released starting from June 2024. Due to its large scale, it will take some time for the dataset to be fully released, considering the uploading speed.

- Will the dataset be distributed under a copyright or other intellectual property (IP) license, and/or under applicable terms of use (ToU)?
  The dataset will be released under the Playground v2.5 Community License license.

- Have any third parties imposed IP-based or other restrictions on the data associated with the instances?
  No.

- Do any export controls or other regulatory restrictions apply to the dataset or to individual instances?
  No.

## D.7 Maintenance

- Who will be supporting/hosting/maintaining the dataset?
  The authors of this paper.

- How can the owner/curator/manager of the dataset be contacted (e.g., email address)?
  Please contact the first author of the paper.

- Is there an erratum?
  No.

- Will the dataset be updated (e.g., to correct labeling errors, add new instances, delete instances)?
  Yes.

- If the dataset relates to people, are there applicable limits on the retention of the data associated with the instances (e.g., were the individuals in question told that their data would be retained for a fixed period of time and then deleted)?
  N/A

- Will older versions of the dataset continue to be supported/hosted/maintained?
  Yes.
- If others want to extend/augment/build on/contribute to the dataset, is there a mechanism for them to do so?
  Please contact the authors of the paper.

