# OpenReview forum: "FiVA: Fine-grained Visual Attribute Dataset for Text-to-Image Diffusion Models"
_NeurIPS.cc/2024/Datasets_and_Benchmarks_Track — NeurIPS 2024 Track Datasets and Benchmarks Poster_

### Official Review · Reviewer_fV3f · 2024-07-22

**Rating:** 7
**Confidence:** 3
**Correctness:** Yes
**Clarity:** Yes

**Review:**

*Quality: The paper is of high quality, presenting a well-structured approach to dataset creation and model training for style transfer and controllability of diffusion models. The experimental results looks robust.
*Clarity: The paper is nicely written, and is easy to follow with detailed descriptions of the methods and comprehensive explanations of the experimental setup and results. Figures and tables are well-organized and easy to understand too.
*Originality: The introduction of a fine-grained visual attribute dataset and the FiVA-Adapter framework for controllable image generation and the approach to utilize multiple images to get visual attributes from for style transfer seem novel.
*Significance: This work addresses the need for more precise control over visual attributes in text-to-image generation models, which is, in my opinion, an important contribution.

**Strengths:**

* Novel large scale visual attribute dataset (consisting 700k pairs).
* The data generation pipeline seems sound.
* FiVA-Adapter allows for the combination of multiple attribute types from different source images,

**Additional Feedback:**

* Grammar error in line 179

**Documentation:**

Yes

**Ethics:**

They use Playground v2.5 to generate their training dataset, which may be generating samples directly from its training set. This is a broadly known ethics issue with generative diffusion models but it is hard to tackle overall. Other than this I think there are no ethical concerns.

**Limitations:**

They addressed the limitations in the supplementary material. I encourage authors to make space and fit limitations into the main paper.

**Opportunities For Improvement:**

* Authors claim to have conducted human validation on 1600 paired images, but did not elaborate how many humans were involved in the analysis and what was the standard deviation between humans etc.
* The reliance on a single text-to-image model (Playground v2.5) for initial data generation may introduce biases or limitations inherent to this model. This may be problematic.
* No dataset/code release. It makes it hard to validate the authors' claims by the community.
* The paper does not discuss the computational resources required for training and running the FiVA-Adapter, which could be a practical concern for broader adoption.
* Please try to fit the limitations in the main paper.

**Relation To Prior Work:**

Yes

**Summary And Contributions:**

The authors generated a Fine-grained Visual Attribute (FIVA) dataset designed for style transfer for text-to-image diffusion models. Overall their framework enhance the customization capabilities of text-to-image models, enabling users to create images with specific visual attributes like lighting, texture, and dynamics. FiVA features a detailed taxonomy of visual attributes and includes 700K high-quality images annotated with these attributes. The authors also propose the FiVA-Adapter framework, which allows for the decoupling and adaptation of visual attributes from multiple source images into a single generated image which is a neat upgrade over the standard IP-Adapter.

---

> ### Author Rebuttal · Authors · 2024-08-17
>
> ### Q1. More details on the human validation process.
>
> Thank you for the constructive suggestion. We would first like to clarify that there are actually 1,400 (200 x 7) image pairs used for human validation, which has been corrected in our supplementary material (L46). In this section, we also briefly introduced the human validation process and visualized the accuracy in Figure S2. We would like to provide more details here.
>
> For the major attributes, a total of 1,400 image pairs were provided for human validation, with 200 pairs for each attribute. Eight well-trained annotators participated in the validation process. Each pair was evaluated by three annotators to assess whether the paired images shared similar visual attributes based on the specific attribute type. We used majority voting to determine the final result. The specific accuracy for each attribute and their average are listed in the first row of the table below, showing a relatively high performance.
>
> We also evaluate the standard deviation between humans as suggested: all eight annotators evaluated the same 350 image pairs (50 pairs per attribute for the 7 attributes), assigning a "0" for misaligned and "1" for well-aligned pairs. The standard deviation, shown in the second row of the table above, reflects a high level of agreement and consistency among the annotators.
>
> | Attribute              | Color | Stroke | Lighting | Dynamic | Focus&DoF | Design | Rhythm | Average |
> | ---------------------- | :---: | :----: | :------: | :-----: | :-------: | :----: | :----: | :-----: |
> | **Accuracy**           | 0.96  |  0.92  |   0.76   |  0.81   |   0.85    |  0.89  |  0.73  |  0.84   |
> | **Standard Variation** | 0.12  |  0.15  |   0.19   |  0.14   |   0.12    |  0.14  |  0.18  |  0.15   |
>
> ---
>
> ### Q2. Reliance on single text-to-image model (Playground-v2.5) may introduces biases.
>
> Thank you for the insightful feedback. We would like to first explain why we use Playground-v2.5. We evaluated several open-sourced state-of-the-art text-to-image models (SDXL-v1.0 [2], PixArt-alpha [4], DALL·E3 [6], Midjourney-v5.2 [7]) on our prompts before formally starting large-scale image generation. However, we found that **(1)** the other open-source models [2,4] showed slightly lower performance in both quality and text alignment; and **(2)** the online products [6,7] rely on APIs for image generation, which is feasible for a small part of image generation to improve data diversity but cannot support large-scale (>100K) data creation due to relatively low efficiency. Playground-v2.5 [1] is open-source, allowing highly efficient batch generation, showing higher quality than SDXL-v1.0 and PixArt-alpha, and comparable performance to the products, which aligns with the user study results in Figure 10 of their technical report [1].
>
> We fully recognize the reviewers' concern regarding the potential risk of data bias by using only one model, and we have noted some newly released powerful models in the last few months, like SD3-medium [3] and Imagen 3 [8]. We plan to generate more samples using different state-of-the-art image generation models [3,5,6,7,8] and update the dataset with higher diversity in model characteristics.
>
> ---
> ### Q3. Dataset and code public availability.
>
> Our dataset is publicly accessible at https://huggingface.co/datasets/FiVA/FiVA/ (Section A.1 of supplementary material).  The repository now includes all image samples, metadata, filtering files, attribute and subject dictionaries, and the dataloader. We also provide detailed documentation on the data structure and file descriptions. For further details and our update plans for the dataset, please refer to our response in **GQ1 of the general rebuttal section** at the top of this page. We have already included dataloader implementation in the link, and we will also public the complete training and inference code for our FiVA-adapter together with the pretrained models on github upon acceptance.
>
> ---
>
> ### Q4. About computational resources required for FiVA-Adapter.
> The model is trained on 8 A100 GPUs for 30 hours. We will discuss the computational resources and more experimental setup details in the final version.
>
> ---
>
> ### Q5. Paper space arrangement and grammar errors.
>
> We will make space and fit the limitations into the main paper, and we will double check to writting to fix grammar errors.
>
> ---
>
> ### Q6. Ethical concern in using Playground-v2.5 model.
>
> Thank you for the thoughtful comment. We acknowledge the concern; however, as the reviewer noted, this is a widely recognized challenge and difficult to fully address. Utilizing powerful pre-trained models like GPT-4 or diffusion models for data generation is a common practice in both AIGC [9] and Large Models [10], which has significantly advanced these fields. We strictly adhere to the models' licenses and take care to avoid any other ethical issues.
>
> ---
>
> **References**
>
> [1] Playground-2.5: Three Insights towards Enhancing Aesthetic Quality in Text-to-Image Generation
>
> [2] SDXL: Improving Latent Diffusion Models for High-Resolution Image Synthesis
>
> [3] Stable Diffusion 3: Scaling Rectified Flow Transformers for High-Resolution Image Synthesis
>
> [4] PixArt-alpha: Fast Training of Diffusion Transformer for Photorealistic Text-to-Image Synthesis
>
> [5] PixArt-sigma: Weak-to-Strong Training of Diffusion Transformer for 4K Text-to-Image Generation
>
> [6] DALL·E3: Improving Image Generation with Better Captions
>
> [7] Midjourney: https://www.midjourney.com/home
>
> [8] Imagen 3: https://deepmind.google/technologies/imagen-3/
>
> [9] InstructPix2Pix: Learning to Follow Image Editing Instructions
>
> [10] LLaVA: Visual Instruction Tuning

---

### Official Review · Reviewer_niWt · 2024-07-23
**Fine-grained Visual Attribute Dataset for Text-to-Image Diffusion Models**

**Rating:** 6
**Confidence:** 5
**Clarity:** The paper is well written, and easy t…

**Review:**

The paper is well written and well motivated. Results cover a good number of ablations that illustrate  the potential for the dataset use and the proposed method performance against different baselines.

The paper main contribution, the FIVA dataset, covers a rich taxonomy of visual attribute annotations, which is a valuable resource for the research community aiming to introduce fine-detail control of image aesthetics appearance.

The human validation section does not describe the number of human experts per image and no support for the conclusion that "the paired images precisely shared the annotated visual attributes". How was this concluded? How was the agreement among annotators evaluated? It only mentions about major attribute categories. What about the subcategories?

The description of this dataset can be improved if included some final overview of its content in the main paper. That is, the current version describes the image generation and filtering processes, but the final version content description can be improved. For instance, (i) it is not clear if the 700k pairs share images across categories or if unique images are used per pair. (ii) It is also not clear the distribution of the 1k subcategories among the initial proposed groups. Are they balanced or how are they distributed? (iii) are the 700k pairs uniformly distributed among the 1k subcategories?

When it comes to the adapter method itself, the novelty is minimal but its strengths rely on the additional level of control introduced. From the formulation presented, it is not clear what is the methodology novelty when compared to the DEADiff. While DEADiff employs two Q-Formers, one for "style" and one for "content," in FIVA adapter, only one is applied for fine grain control over specific visual attributes. By removing the content adapter, FIVA lacks some of the properties of the original method (DEADiff follows the target subject better). Thus the major impact of the method is introducing fine detail aesthetic control. My main question to the authors is why the second Q-former was removed from the original method, instead of combining content with fine detail control of the attributes. Were there any attempts to combine both? If yes, any side effects noted? If not, please justify.

In summary, the impact of the paper is reduced due to (i) the proposed dataset not being public, (ii) lack of description of its human validation process (iii) lack of novelty on the proposed adapter method.

**Strengths:**

The proposed dataset differentiates from existing datasets for image generation by providing fine-grain visual/aesthetics attributes annotations organized in a taxonomy of 1k specific subcategories.

Additionally to the dataset, they also present an adaption method. Existing works for controlled generation enforce consistent style between reference and generated images in a more implicit abstract way, in which the definition of style may be inconsistent or hard to control. The propose a fine-grained visual attribute adapter aims to decouple and adapt visual attributes from one or more source images with fine grain control into generated images, enhancing the specificity and applicability of the generated content.

Authors present experiments across a variety of attribute types on both synthetic and real-world test sets. The results show that the method outperforms baselines in terms of precise controllability in attribute extraction, high textual alignment regarding the target prompt, and the flexibility to combine different attributes.

**Additional Feedback:**

In order to increase the impact of the proposed paper, it must contain a straightforward recipe for replicating the proposed dataset.

**Correctness:**

The paper claims are reasonable. The initial steps of the construction of the dataset are well described, but not its human evaluation filtering and final composition.
There is not a new benchmark associated with the paper.

**Documentation:**

(from the checklist section)
The code and the data are proprietary
The license to the code and datasets is a broken reference ("... [Yes] See Section ??.")

"385 Did you include the license to the code and datasets? [Yes] See Section ??.
386 Did you include the license to the code and datasets? [No] The code and the data are
387 proprietary."

**Limitations:**

The impact of the FIVA dataset may be limited by two complementary aspects: size and the use of generated images. Given the large number of visual attributes covered (1k subcategories) the size of the dataset (700k pairs) may be relatively small.  The lack of description of the final dataset, of the human validation process, and the fact that the paper does not present a  public dataset, limits its replicability and the paper extenal impact.

The FIVA-adapter method is limited by being highly dependent on the FiVA-dataset as it relies on the specifically designed fine-grained annotations, and possibly also limited by the removal of the second q-former responsible for content conditioning. The extension of the method contribution as an adaptation method in other settings is not clear, given the similarity with the baseline adapter methodology (DEADiff)

**Opportunities For Improvement:**

- Describe the contents of the final FIVA dataset such as the distribution of images per group and subcategory

- Describe the motivation for changing the baseline method DEADiff and the removal of one of their Q-former.

- Describe the human validation process, otherwise the dataset is not replicable

**Relation To Prior Work:**

The paper presents a good set of previous work and discusses the dataset and method contributions in relation to them.

**Summary And Contributions:**

The authors claim to decompose aesthetics of a picture into specific visual attributes and introduce FiVA (Fine-grained Visual Attribute) dataset, a dataset containing 700k image pairs annotated with fine-grained visual attributes. Each image pair is associated with the same fine grain aesthetic attribute.

The dataset is built by using 2D generative models with carefully designed prompts covering different visual attributes and a filtering process involving LLMs and human validation. Initially they generate prompts containing various visual attributes in scale, and then we use state-of-the-art text-to-image models to generate images with the corresponding prompts. The generated images with the same attribute are considered to be positive pairs that are further filtered for consistency using GPT-4. A human validation of visual attributes consistency filtering is applied selecting random pairs for each category, sustaining the analysis of the dataset annotation precision.

Additionally, they propose FiVA-Adapter, a novel visual prompt adapter designed to utilize FiVA. This adapter leverages a Q-former module to extract attribute-specific image features from reference images and incorporates them into the generation process through a multi-image dual cross-attention module.
Experimental results demonstrate that FiVA-Adapter, trained on FiVA, outperforms existing methods in accurately transferring specific attributes to the generated images while maintaining subject fidelity and offering greater flexibility in combining multiple attributes.

Contributions:
- Introduce a taxonomy of visual attributes that subdivides main attributes groups in a total of 1k subcategories covering a large range of applications and visual domains.
- FiVA dataset: A large-scale dataset of 700k image pairs with fine-grained visual attribute annotations, enabling research on controllable image generation.
- FiVA-Adapter: A novel visual prompt adapter specifically designed to extract and utilize fine-grained visual attributes from reference images during text-to-image generation.
- Ablations: quantitative and qualitative evaluations demonstrate the effectiveness of the proposed approach in controlling various visual attributes, surpassing state-of-the-art methods in terms of attribute accuracy, subject fidelity, and flexibility.

---

> ### Author Rebuttal · Authors · 2024-08-17
>
> ### Q1. Dataset public availability.
>
> Our dataset is now publicly available at https://huggingface.co/datasets/FiVA/FiVA/ . Please refer to the **GQ1 in the general rebuttal section** at the top of this page for more details.
>
> ---
>
> ### Q2. More details on the human validation process.
>
> Thank you for the valuable comment. We introduced the human validation process in Section B (L46-51 and Figure S2) of the supplementary material and will provide further details here.
>
> **(1)** For the major attributes, 1,400 image pairs (200 per attribute) were evaluated by eight trained annotators, with majority voting used to determine attribute similarity. The accuracy for each attribute and their average, shown in the first row of the table, indicates precise alignment. Due to the extremely high cost of validating 1K subattributes separately by human annotators, we uniformly sample examples from each major attribute for evaluation.
>
> Annotator agreement was also tested on 350 image pairs (50 per attribute), with standard deviations in the second row of the table indicating high consistency.
>
> | Attribute              | Color | Stroke | Lighting | Dynamic | Focus&DoF | Design | Rhythm | Average |
> | - | :-: | :-: | :-: | :-: | :-: | :-: | :-: | :-: |
> | **Accuracy**           | 0.96  |  0.92  |   0.76   |  0.81   |   0.85    |  0.89  |  0.73  |  0.84  |
> | **Standard Variation** | 0.12  |  0.15  |   0.19   |  0.14   |   0.12    |  0.14  |  0.18  |  0.15  |
>
> ---
>
> ### Q3. Improve the description of the dataset.
>
> Thank you for the valuable suggestions. We clarify some details below and will further improve the content description in the final version.
>
> **(i)** The dataset includes 700K (now increased to 1.04M) unique image samples as mentioned in the abstract (we will correct the typo in the introduction).
>
> **(ii)** The distribution of subcategories is shown in the table below:
>
> | Attribute              | Color | Stroke | Lighting | Dynamic | Focus&DoF | Design | Rhythm |
> | - | :-: | :-: | :-: | :-: | :-: | :-: | :-: |
> | Sub-attribute number   |  311  |  271   |   257    |   84    |    77     |   -    |   -    |
> | Description set number |  311  |  271   |    40    |   12    |    11     |  600   |  300   |
>
> **Description set number** refers to the number of prompt/keyword groups used to describe a specific visual attribute, while **Sub-attribute number** represents the actual distinct visual forms within the major attribute. We provide examples in **Figure R1 of the rebuttal PDF attached** in this section to illustrate why they differ.  'Design' and 'Rhythm' are not included in the subcategory count, as explained in Section B of the supplementary material, since these complex attributes are described with long descriptions rather than being categorized as subattributes.
>
>  **(iii)** Image sample distribution acorss major attributes is shown in the table below. We report the number of images containing each attribute. The total does not equal 1.04M (the current dataset size on Hugging Face) because some images contain multiple attributes. The number of images with 1, 2, or 3 attributes are 407K, 461K, and 171K, respectively.
>
> The data distribution is not fully balanced, as some attributes (e.g., color, stroke, lighting) are more general and can be widely applied to many subjects, while others are more limited to specific subject sets. In real-world applications, the frequency and demand for different attributes are also inherently imbalanced. To reduce bias, we adopted a repeat factor sampler [1], which helps balance the image pair sampling ratios for each attribute, as shown in the second row of the table.
>
> | Attribute               | Color | Stroke | Lighting | Dynamic | Focus&DoF | Design | Rhythm | Total image number |
> | - | :-: | :-: | :-: | :-: | :-: | :-: | :-: | :-: |
> | Individual image number | 429K  |  301K  |   370K   |   89K   |   107K    |  66K   |  52K   |       1.04M        |
> | Image-pair ratio        | 26% | 15% | 16% | 11% | 12% | 11% | 9% |         -          |
>
> ---
>
> ### Q4. About the data scale.
>
> Our dataset has now grown to 1.04M images (publicly available on Hugging Face), and we plan to generate additional samples using various state-of-the-art models based on the reviewers' suggestions to further enhance its quality. For more details, please refer to **GQ2 in the general rebuttal section** at the top of this page.
>
> ---
>
> ### Q5. The reason for not applying the second Q-former in the FiVA-Adapter. Explain the extension of FiVA-Adapter.
>
> Thank you for the thoughtful question. We would like to justify the reason behind our design as below:
>
> 1. Our motivation is to facilitate applications that apply specific fine-grained visual attributes (rather than a simple 'style') from different reference images to a target image. For DEADiff, it is an innovative work that disentangled the 'content' and 'style' of images with dual Q-formers, making the Stylization Diffusion Model follow the 'style' of reference images better. In summary, we focus on fine-grained control of visual attributes. And disentangling the 'content' and 'style' is not our goal.
> 2. As a Dataset Track paper, we mainly focus on constructing our FiVA dataset for fine-grained visual attributes. The FiVA-Adapter is a baseline framework to verify the FiVA dataset. We wish it to be simple and easy to be extended with other designs. Therefore, to make the overall architecture and training pipeline clean, FiVA-Adapter doesn't adopt the second 'content' Q-former proposed in DEADiff.
>
> As for the extensions of FiVA-Adapter: First, it can be further equipped with advanced designs for better results, e.g., the content Q-Former in DEADiff. Second, to make free-form natural language control (rather than attribute names), FiVA-Adapter can be integrated with large multimodal models which handle text and image conditions better.
>
> ---
>
> **References**
>
> [1] LVIS: A Dataset for Large Vocabulary Instance Segmentation

---

### Official Review · Reviewer_cCrU · 2024-07-25
**Dataset for text-to-image generation with fine-grained visual adttributes**

**Rating:** 6
**Confidence:** 5
**Correctness:** Yes
**Clarity:** This paper is clearly written.

**Review:**

- The proposed FiVA dataset annotated with fine-grained visual attributes enables the fine-grained control in text-to-image generation.
- To create the dataset, GPT-4 is first used to associate visual attributes with subjects. Then, prompts are created by combining one subject with various numbers of attributes. With the initial prompts, four images were created for each prompt using the playground-v2.5 model. A cleaning and filtering process matched the text to the image and created a reliable dataset of images with specific visual attributes.
- Fine-grained Visual Attribute Adapter (FiVA-Adapter) was proposed for controllable image generation that reflects the attritubes.
- The detailed explanation of the dataset composition and creation process is omitted, including the process of determining attributes for the dataset and the human-validation process to verify the reliability and versatility of the dataset. Since these procedures play a crucial role in ensuring the quality and applicability of the dataset, more detailed explanations should be included in the manuscript.

**Strengths:**

- The motivation to create the fine-grained visual attribute dataset is reasonable.
- FiVA-Adapter outperforms the existing text-to-image generation models when transferring attributes.

**Additional Feedback:**

None.

**Documentation:**

Unfortunately, the paper does not contain a link to access the dataset and there is no information about the release of the data and code attached to the paper.

**Ethics:**

No. There are no ethical concerns.

**Limitations:**

- The number of data
- Only the playground-2.5 model was used to create the images according to the given prompt.

**Opportunities For Improvement:**

- The dataset consisting of 700K samples may be insufficient to be considered a large dataset. Previously proposed datasets in this field typically comprise millions of data samples. This discrepancy in scale could potentially limit the robustness and generalizability of the model trained on this dataset compared to those trained on larger, more comprehensive datasets.
- The exact number of images for each attribute is not clearly described. If the proportion of data for each attribute is not relatively balanced, this could potentially lead to attribute-specific biases in the dataset. This information is also crucial for assessing the generalizability of the trained models using this dataset.
- The images corresponding to the prompts were generated using playground-v2.5 only. Despite the existence of a variety of text-image generation models, the use of only one model to create the dataset implies that the characteristics of the dataset may be biased towards the dataset trained with playground-v2.5 and the performance of that model.

**Relation To Prior Work:**

The authors clearly described the difference between the proposed dataset and the existing datasets for text-to-image generation.

**Summary And Contributions:**

The paper presents a new text-to-image generation dataset containing 700,000 high-quality images with annotated visual attributes. The proposed FiVA dataset decomposes image aesthetics into distinct visual attributes such as lighting, texture, dynamics, etc. In addition, they developed the FiVA adapter framework that utilizes this dataset to selectively apply specific visual attributes from multiple source images to generated images.

---

> ### Author Rebuttal · Authors · 2024-08-17
>
> ### Q1.  Detailed explanation of the dataset composition and creation process.
>
> #### **1) More details for the process of determining attributes.**
>
> Thank you for the valuable feedback. As this is a novel problem in image generation, we found no prior work directly available for reference. Thus, we relied on a range of sources, including notable photography books (e.g., *The Photographer's Eye* by Michael Freeman) and expert consultations, to define the visual attribute dimensions in our paper.
>
> To establish subcategories for each dimension, we first referenced examples from professional texts, then used GPT-4 to generate additional entries. We further manually filter out any redundant or unreasonable suggestions. Additionally, we evaluated the visual consistency of each subattribute and implemented a range-sensitive filter (Section 3 & Section B) to further improve the visual attribute alignment between image pairs.
>
> #### **2) More details for the human validation process.**
> We briefly introduced the human validation process in Section B (L46-51 and Figure S2) of the supplementary material and will provide more details here. For validation, 1,400 image pairs (200 per attribute) were evaluated by eight well-trained annotators. Each pair was assessed by three annotators based on attribute similarity, with majority voting determining the final result. Accuracy for each attribute, along with their average, is listed in the first row of the table, showing a relatively high accuracy.
>
> Additionally, we tested human agreement by having all eight annotators evaluate 350 image pairs (50 per attribute). The standard deviation is provided in the second row of the table, indicating a high level of consistency among them.
>
> | Attribute              | Color | Stroke | Lighting | Dynamic | Focus&DoF | Design | Rhythm | Average |
> | - | :-: | :-: | :-: | :-: | :-: | :-: | :-: | :-: |
> | **Accuracy**           | 0.96  |  0.92  |   0.76   |  0.81   |   0.85    |  0.89  |  0.73  |  0.84  |
> | **Standard Variation** | 0.12  |  0.15  |   0.19   |  0.14   |   0.12    |  0.14  |  0.18  |  0.15  |
>
> ---
>
> ### Q2. About data scale.
>
> Thank you for the thoughtful feedback. Our dataset has now grown to 1.04M images (publicly available on Hugging Face), and we plan to generate additional samples using various state-of-the-art models based on the reviewers' suggestions to further enhance its quality. For more details, please refer to **GQ2 in the general rebuttal section** at the top of this page.
>
> ---
>
> ### Q3 About data distribution.
> In the table below, we report the number of images containing each attribute. The total does not equal 1.04M (current dataset size on Hugging Face) because some images contain multiple attributes. The number of images with 1, 2, or 3 attributes are 407K, 461K, and 171K, respectively.
>
> The data distribution is not fully balanced, as some attributes (e.g., color, stroke, lighting) are more general and can be largely adapted to many subjects, while others are more limited to specific subject sets. In real-world applications, the frequency and demand for different attributes are also inherently imbalanced. To reduce bias, we adopted a repeat factor sampler [8], which largely helps balance the image pair sampling ratios for each attribute. This is reflected in the second row of the table.
>
> | Attribute               | Color | Stroke | Lighting | Dynamic | Focus&DoF | Design | Rhythm | Total image number |
> | - | :-: | :-: | :-: | :-: | :-: | :-: | :-: | :-: |
> | Individual image number | 429K  |  301K  |   370K   |   89K   |   107K    |  66K   |  52K   |       1.04M        |
> | Image-pair ratio        | 26% | 15% | 16% | 11% | 12% | 11% | 9% |         -          |
>
> ---
>
> ### Q4. Bias to Playground-2.5.
> Thank you for the insightful feedback. We chose Playground-v2.5 after evaluating several state-of-the-art text-to-image models (SDXL-v1.0 [2], PixArt-alpha [4], DALL·E3 [6], Midjourney-v5.2 [7]). **(1)** Open-source models [2,4] showed slightly lower performance in quality and text alignment, and **(2)** DALL·E3 and Midjourney rely on APIs, which are suitable for small-scale generation but inefficient for large-scale (>100K) data creation. Playground-v2.5 [1] is open-source, supports efficient batch generation, showing higher quality than other open-source models and comparable performance to the products, which aligns with the user study results in Figure 10 of their technical report [1].
>
> We fully recognize the reviewers' concern about potential bias from using a single model, and we have noted some newly released powerful models like SD3-medium [9] and Imagen 3 [10]. We are planning to generate more samples using newer models to further enhance the dataset's diversity in models.
>
> ___
>
> ### Q5. Dataset link and documentation.
> Our dataset is now publicly available at https://huggingface.co/datasets/FiVA/FiVA/ . Please refer to the **GQ1 at the general rebuttal section** at the top of this page for more details.
>
> ---
>
> **References**
>
> [1] Playground-2.5: Three Insights towards Enhancing Aesthetic Quality in Text-to-Image Generation
>
> [2] SDXL: Improving Latent Diffusion Models for High-Resolution Image Synthesis
>
> [3] Stable Diffusion 3: Scaling Rectified Flow Transformers for High-Resolution Image Synthesis
>
> [4] PixArt-alpha: Fast Training of Diffusion Transformer for Photorealistic Text-to-Image Synthesis
>
> [5] PixArt-sigma: Weak-to-Strong Training of Diffusion Transformer for 4K Text-to-Image Generation
>
> [6] DALL·E3: Improving Image Generation with Better Captions
>
> [7] Midjourney: https://www.midjourney.com/home
>
> [8] LVIS: A Dataset for Large Vocabulary Instance Segmentation
>
> [9] Stable Diffusion 3: Scaling Rectified Flow Transformers for High-Resolution Image Synthesis
>
> [10] Imagen 3: https://deepmind.google/technologies/imagen-3/

---

### Author Rebuttal · Authors · 2024-08-17

We sincerely thank all the reviewers for their insightful and constructive comments. We are grateful and encouraged that reviewers find our paper *novel and of high quality* **[fV3f]**, and that our FiVA-adapter *enhances the specificity and applicability of the generated content* **[niWt]**, *outperforming the existing text-to-image generation models* **[cCrU]** *in terms of attribute accuracy, subject fidelity, and flexibility* **[niWt]**. It *addresses the need for more precise control over visual attributes in text-to-image generation models, being an important contribution* **[fV3f]**. Most reviewers point out that our paper is well-written and well-motivated **[cCrU, niWt, fV3f]**. We are grateful for all these positive feedbacks.

We have provided detailed responses to each reviewer separately regarding the concerns and opportunities for improvement. Below, we provide information on how to publicly access and utilize our dataset, along with future update plans and a discussion on data scale, which we believe will be of interest to all reviewers.

---

### **GQ1. Dataset public availability.**
Our dataset is publicly available on https://huggingface.co/datasets/FiVA/FiVA/ as introduced in Supplementary Material - Section A.1 . The repository now includes the following stuff:

- **1.04M** unique image samples.
- Metadata files and a detailed document.
- Taxonomy file for all the visual attributes and subjects.
- Code for dataset and dataloader module to use the data.
- A README that makes a clear explanation on all the files above.

**Updating Plans**

1. Involve more image samples from different state-of-the-art image generation models.
2. We will public the complete training and inference code for our FiVA-adapter together with the pretrained models on github upon acceptance.

---

### **GQ2. About data scale.**

Thank you for the thoughtful feedback. Compared to datasets built from large-scale existing images, it is indeed challenging to achieve a large data scale when the diffusion model-based generation process is part of the pipeline. This is primarily due to the heavy computational costs and limited resources we have. Similar challenges have been noted in previous works: InstructPix2Pix [1] generated only 450K (0.45M) images, and DEADiff [2] produced 1.06M. We initially generated 700K (0.7M) images, which has now increased to **1.04M**.

Given that the dataset addresses a specific yet important task of visual attribute learning, we find that the current data scale can already support robust performance based on extensive experiments across synthetic and real-world datasets. However, we acknowledge room for improvement. Following the reviewers' suggestions, we plan to generate additional samples using various state-of-the-art models, such as SD3-medium [3], PixArt-sigma [4], DALL·E3 API [5], Midjourney API [6], and Imagen [7]. In the process, we will ensure full compliance with the licenses and terms of use for each model's generated images. We will update the public dataset with these new data, addressing concerns regarding both data bias and scale.

---

**References**

[1] InstructPix2Pix: Learning to Follow Image Editing Instructions

[2] DEADiff: An Efficient Stylization Diffusion Model with Disentangled Representations

[3] Stable Diffusion 3: Scaling Rectified Flow Transformers for High-Resolution Image Synthesis

[4] PixArt-sigma: Weak-to-Strong Training of Diffusion Transformer for 4K Text-to-Image Generation

[5] DALL·E3: Improving Image Generation with Better Captions

[6] Midjourney: https://www.midjourney.com/home

[7] Imagen 3: https://deepmind.google/technologies/imagen-3/

---

### Comment · Area_Chair_orzz · 2024-08-30
**Urgent: Reminder to Review Author Rebuttals and Engage in Discussion**

Dear Reviewers,

We are just two days away from the end of the discussion period. The authors have provided their rebuttals to the reviews. It is very important that you review the rebuttal and other reviews, and engage in the conversation or respond if the rebuttal addresses your concerns.

Thank you.

---

> ### Comment · Area_Chair_orzz · 2024-08-31
> **Urgent: Final Reminder to Review Author Rebuttals**
>
> Urgent: Final Reminder to Review Author Rebuttals

---

### Decision · Program_Chairs · 2024-09-26

**Decision:**

Accept (Poster)

**Comment:**

This paper defines a comprehensive taxonomy of visual attributes in images. It then introduces the FiVA dataset, which contains 700K high-quality synthesized image pairs, where each pair shares one fine-grained visual attribute. Using this dataset, the paper presents the FiVA-adapter framework, which can extract and combine fine-grained visual attributes from multiple images for selective transfer to a target generated image. Experiments demonstrate that the resulting model surpasses existing style transfer models in controlling fine-grained attributes during image generation.

All three reviewers leaned towards accepting the paper. They appreciated that it introduces a large, high-quality dataset annotated with a comprehensive taxonomy of fine-grained attributes, differentiating it from existing text-to-image datasets. They also noted that the data generation pipeline is sound and that the paper provides sufficient verification to ensure data quality. Furthermore, the experiments demonstrate significant improvements in the controllability of trained models compared to the baselines. During the rebuttal, the authors addressed questions regarding the human verification process and the distribution of the dataset with respect to attributes, clarified some of their design choices, and acknowledged certain limitations of the pipeline (e.g., bias due to generation with only one model). The AC thinks that the dataset generation pipeline is novel and sound, and that this dataset would be highly beneficial to researchers working on controllable image generation and fine-grained controls, and therefore recommends accepting the paper.